# Outcomes and Adverse Effects of Voretigene Neparvovec Treatment for Biallelic *RPE65*-Mediated Inherited Retinal Dystrophies in a Cohort of Patients from a Single Center

**DOI:** 10.3390/biom13101484

**Published:** 2023-10-05

**Authors:** Peter Kiraly, Charles L. Cottriall, Laura J. Taylor, Jasleen K. Jolly, Jasmina Cehajic-Kapetanovic, Imran H. Yusuf, Cristina Martinez-Fernandez de la Camara, Morag Shanks, Susan M. Downes, Robert E. MacLaren, M. Dominik Fischer

**Affiliations:** 1Oxford Eye Hospital, Oxford University Hospitals NHS Foundation Trust, Oxford OX3 9DU, UK; peter.kiraly20@gmail.com (P.K.); jasmina.kapetanovic@ouh.nhs.uk (J.C.-K.); imran.yusuf@eye.ox.ac.uk (I.H.Y.); cristina.martinez@ouh.nhs.uk (C.M.-F.d.l.C.); susan.downes@ouh.nhs.uk (S.M.D.); enquiries@eye.ox.ac.uk (R.E.M.); 2Nuffield Laboratory of Ophthalmology, University of Oxford, Oxford OX3 9DU, UK; jasleen.jolly@aru.ac.uk; 3Vision and Eye Research Institute, Anglia Ruskin University, Cambridge CB1 1PT, UK; 4Oxford Regional Genetics Laboratories, Oxford University Hospitals NHS Foundation Trust, Oxford OX3 9DU, UK; morag.shanks@ouh.nhs.uk; 5Centre for Ophthalmology, University Hospital Tubingen, 72076 Tubingen, Germany

**Keywords:** voretigene neparvovec, gene therapy, *RPE65*-mediated inherited retinal dystrophies, IRD, functional outcomes, adverse effects, retinal atrophy, high IOP

## Abstract

Our study evaluated the morphological and functional outcomes, and the side effects, of voretigene neparvovec (VN) gene therapy for RPE65-mediated inherited retinal dystrophies (IRDs) in 12 eyes (six patients) at the Oxford Eye Hospital with a mean follow-up duration of 8.2 (range 1–12) months. All patients reported a subjective vision improvement 1 month after gene therapy. Best-corrected visual acuity (BCVA) remained stable (baseline: 1.28 (±0.71) vs. last follow-up: 1.46 (±0.60); *p* = 0.25). Average white Full-Field Stimulus Testing (FST) showed a trend towards improvement (baseline: −4.41 (±10.62) dB vs. last follow-up: −11.98 (±13.83) dB; *p* = 0.18). No changes in central retinal thickness or macular volume were observed. The side effects included mild intraocular inflammation (two eyes) and cataracts (four eyes). Retinal atrophy occurred in 10 eyes (eight mild, two severe) but did not impact FST measurements during the follow-up period. Increased intraocular pressure (IOP) was noted in three patients (six eyes); four eyes (two patients) required glaucoma surgery. The overall safety and effectiveness of VN treatment in our cohort align with previous VN clinical trials, except for the higher occurrence of retinal atrophy and increased IOP in our cohort. This suggests that raised IOP and retinal atrophy may be more common than previously reported.

## 1. Introduction

Inherited retinal dystrophies (IRDs) represent a significant cause of blindness among the working-age population [1]. IRDs are known for their phenotypic and genotypic heterogeneity, with over 280 genes being associated with various forms of IRD [2]. Biallelic mutations in *RPE65*, which encodes the isomerase of the retinoid cycle, have been associated with retinitis pigmentosa type 20 (RP) and Leber congenital amaurosis type 2 (LCA). *RPE65* mutations are responsible for approximately 2–16% of mutations observed in patients with LCA, and they account for 1–2.7% of mutations in patients with autosomal recessive RP [3]. While the clinical presentation (e.g., age at onset, pattern, rate of progression), and hence, diagnostic labels (RP, LCA, early-onset retinal dystrophy (EORD), etc.) can vary between patients, the underlying disease caused by *RPE65* mutations eventually leads to complete blindness if left untreated. Until recently, no treatment options were available and supportive measures were considered the best management for these patients. After phase I/II and phase III studies demonstrated the safety and efficacy of gene replacement therapy using voretigene neparvovec (VN) in patients with *RPE65*-mediated IRDs, the US Food and Drug Administration (FDA) and the European Medicines Agency (EMA) granted their approval for the first ocular gene therapy [4,5].

In the pivotal phase III study, which involved 21 patients receiving gene therapy with VN and a control group of 10 patients, the primary efficacy endpoint was based on multi-luminance mobility testing (MLMT), which demonstrated a statistically significant improvement in the treated group after one year [5]. Furthermore, mean white Full-Field Stimulus Testing (FST) revealed a significant improvement in the intervention group after one month, and this improvement was sustained and remained stable at the one-year follow-up [5]. Treatment with VN resulted in nearly a twofold increase in the average sum of total degrees when assessed using the Goldmann visual field (GVF) [5]. On the other hand, the intervention group did not demonstrate a statistically significant improvement in best-corrected visual acuity (BCVA) [5]. The subsequent long-term results of this study revealed sustained improvements in MLMT, FST and GVF for the full duration of the follow-up reported (up to 4 years) [6]. Smaller retrospective studies confirmed improvements following VN treatment in behavioral changes, including improved mobility and reduced dependence on assistive devices, FST in the majority of patients and improved BCVA and retinal sensitivity in some patients [7,8,9,10,11,12]. After VN treatment, two eyes showed partial recovery in light-adapted 30 Hz flicker electroretinography (ERG), as documented in one of the studies [11]. Significant improvements following the VN treatment were primarily observed in younger pediatric patients with *RPE65* mutations [9,11,12]. However, the older population with a reduced number of viable retinal cells showed limited improvement [9]. A case report of a 15-year-old patient with an *RPE65* mutation who underwent VN treatment demonstrated improvements in foveal retinal morphology that were partially correlated with improvements in BCVA [13]. Another case report described the restoration of the bisretinoid fluorescent signal, as observed on quantitative autofluorescence (AF), six years following the VN treatment [14]. 

In the pivotal study, mild ocular side effects were reported during the one-year observation period following the VN treatment [5]. These included a transient increase in intraocular pressure (20%), mild intraocular inflammation (10%), cataract formation (15%), iatrogenic retinal tears (10%), macular holes (5%) and epiretinal membranes (5%). These side effects were mostly associated with the surgical procedures involved, such as vitrectomy and subretinal injection, and not related to the vector used [5]. The adverse effect profile was similar in another gene therapy study that involved vitrectomy and subretinal injection, but with a different vector [15]. This supports the assertion that the reported side effects are linked to the vitrectomy and subretinal injection procedures rather than the vector itself. A case report demonstrated macular fold development, resulting in the irreversible loss of BCVA one day after a vitrectomy with subretinal VN application [16]. The development of an asymptomatic iatrogenic choroidal neovascular complex (CNV) was also reported as a surgical adverse effect following gene augmentation therapy for *RPE65*-mediated IRDs [17]. 

In 2022, Gange et al. first described the development of progressive retinal atrophy following treatment with VN [18]. In another retrospective study, retinal atrophic changes were observed in all 13 eyes that underwent VN treatment [10]. In 2023, Lopez et al. reported the formation of subretinal deposits in three young patients one week after subretinal VN application [19]. All patients demonstrated similar improvements in visual function after the treatment, consistent with the findings reported in previous studies [5,6,19]. The authors hypothesized that the formation of subretinal deposits could be attributed to an immune response triggered by the adeno-associated viral vector [19].

In our study, we observed BCVA, FST, central retinal thickness and macular volume outcomes and adverse effects of VN treatment for biallelic *RPE65*-mediated IRDs in a cohort of patients from the Oxford Eye Hospital. Real-world studies like ours are crucial for confirming drug effectiveness across heterogeneous populations and offering insights into long-term safety, especially concerning rare adverse events. Moreover, they provide information about usage trends and their associated health and economic impacts [20]. We highlight that patients with more advanced disease and a limited number of viable retinal cells remaining experience less benefit. We confirm that retinal atrophy is an important side effect and describe a case of retinal atrophy affecting the fovea. We also report that glaucoma surgery (goniotomy or tube surgery) was required to control IOP after VN treatment in two patients and that glaucomatous changes developed in one patient. 

## 2. Methods

### 2.1. Study Design and Participants

Our retrospective study included 6 patients (12 eyes) with biallelic *RPE65*-mediated IRDs treated with bilateral VN at the Oxford Eye Hospital, Oxford University Hospitals NHS Foundation Trust. The study adhered to the tenets of the Declaration of Helsinki and was approved by the West Midlands—Edgbaston Research Ethics Committee (reference 20/WM/0176). To be included in the study, participants had to meet the following three criteria (as per label): a clinical diagnosis of inherited retinal disease, the presence of biallelic *RPE65* mutations, and sufficient viable retinal cells. All three criteria were reviewed by a multidisciplinary team (MDT), confirming the indication for treatment. After discussing the pros/cons of treatment with VN, patients provided informed consent for the treatment. Separate informed consent was obtained for participation in the post-approval safety study PERCEIVE, sponsored by Novartis [21].

### 2.2. Procedures

Patients were treated by two vitreoretinal surgeons (MDF and REM) who have extensive experience in the application of subretinal gene therapy, as described by Russell et al. [5]. After performing a standard core vitrectomy, triamcinolone acetonide (Kenalog, Bristol Myers Squibb, Princeton, NJ, USA) was utilized to visualize the remaining vitreous as these IRD patients usually show significant vitreoschisis during posterior vitreous detachment. A total of 300 µL of 1.5 × 10^11^ VN solution was slowly injected using a foot-pedal-operated injection system, which had a maximum injection pressure of 10 psi, over a duration of 30 to 60 s. The VN solution was directly injected into the subretinal space without the use of a subretinal pre-bleb.

In all patients, foveal detachment was confirmed using intraoperative OCT (Zeiss RESCAN 700, OPMI Lumera, Carl Zeiss, Oberkochen, Germany). Three eyes had one bleb, one eye had two blebs, five eyes had three blebs, and two eyes had four blebs raised. Patients began taking oral prednisolone as per the packaging instructions: 1 mg/kg/day (max 40 mg/d) starting three days prior to surgery, which they continued for a duration of seven days. This was followed by five days of 0.5 mg/kg/day (max 20 mg/d), followed by 0.5 mg/kg prednisolone (max 20 mg/d) every second day over five days. If the second eye was treated within 14 days of the first, then the higher (pre-operative) dosing schedule from the relevant day was followed instead. The second eye surgery was performed one week after the first eye surgery in five patients, while in one patient, the second eye surgery was performed after one year (due to the COVID-19 pandemic).

Patients underwent baseline examinations prior to surgery, and post-surgery follow-up examinations were undertaken at 1 month, 3 months, 6 months, and 12 months after the first eye surgery. Ophthalmological examinations included slit lamp examination via fundoscopy, the measurement of BCVA and IOP, optical coherence tomography (OCT), fundus autofluorescence (FAF), fundus wide-angle imaging and FST.

### 2.3. Best-Corrected Visual Acuity

BCVA was assessed using optimal refraction correction and following the standard protocol with the Early Treatment Diabetic Retinopathy Study (ETDRS) visual acuity chart (Precision Vision, Woodstock, IL, USA) [22]. For patients with BCVA too poor to obtain measurable responses using the ETDRS chart, the Berkeley Rudimentary Vision Test (BRVT) was employed as an alternative [23]. For statistical analysis, logarithm of the minimum angle of resolution (LogMAR) values were utilized.

### 2.4. Multimodal Imaging

Spectral domain (SD)-OCT and 488 nm AF images were obtained using the Spectralis imaging platform (Heidelberg Engineering, Inc., Heidelberg, Germany). Volume OCT scans were acquired, comprising 19 horizontal B-scans with fixation at the fovea. The macular volume was determined within the 3 mm area defined by the ETDRS grid. The central retinal thickness was automatically determined by the OCT software (Heidelberg Eye Explorer 2.0) within a 1 mm radius around the foveola. In cases of inappropriate automatic segmentation due to advanced retinal pathology, manual segmentation was performed by a medical retina specialist. Pseudo-color wide-field fundus images were obtained using Optos (Optomap P200; Optos plc, Dunfermline, UK).

### 2.5. Full-Field Stimulus Testing

Dark-adapted FST was performed following pupil dilation to assess retinal sensitivity to 6500 K 4 s white flashes with an interstimulus interval of 5 s following 45 min dark adaptation using the Diagnosys Espion system (Diagnosys LLC, Cambridge, UK). A value of 0 dB was set to 0.01 cd/m^2^. The 2-button box was used with audible cues in a 2-force choice algorithm using a staircase and Weibull fit function with a 50% probability of detection in order to calculate the threshold. The eye not being tested was double-patched to prevent light leakage. Each eye was tested four times with a period of 5 min of re-adaptation in between each test. The first test was discarded as a learning test and subsequent tests were averaged for the final result. The right eye was always tested first.

### 2.6. Statistical Analysis

To assess the statistical significance between variables before surgery and at the last follow-up, the Wilcoxon signed-rank test was employed. A two-sided *p*-value threshold of 0.05 was used to determine statistical significance. The analysis was conducted using IBM SPSS Statistics for Windows, version 28 (IBM Corp., Armonk, NY, USA). 

## 3. Results

The average age of patients at the time of surgery was 36.3 years (range: 18–49 years); the group comprised five males and one female. All patients underwent molecular genetic analysis, which confirmed the presence of biallelic *RPE65* mutations. In all patients, a molecular genetic analysis using the RP 111 Gene Panel was performed to exclude other mutations causing RP and LCA phenotypes. Clinically, four patients were diagnosed with RP and two patients with LCA type 2. Biallelic *RPE65* mutations were homozygous in four patients and compound heterozygous in two patients (Table 1).

Subjective vision improvement was reported by all patients one month after the gene therapy. Patient 1 (P1) reported improved visual function in low-light conditions. Patient 2 (P2), who had received gene therapy in the right eye a year prior to the left eye, reported an increased peripheral visual field in the right eye and improved light sensitivity in the left eye. Patient 3 (P3) described a significant improvement in eyesight, particularly noticeable under dim lighting. P3 mentioned using lower brightness settings on her phone and being able to see parked cars on the street in the dark after the gene therapy. Patients 4 (P4), 5 (P5) and 6 (P6) all reported improved sensitivity to light.

The average BCVA, converted to LogMAR, remained relatively unchanged after VN treatment, with a baseline value of 1.28 (±0.71) compared to 1.46 (±0.60) at the last follow-up (*p* = 0.25). The average FST with white light showed improvement from −4.41 (±10.62) dB at baseline to −11.98 (±13.83) dB at the final follow-up. However, this improvement was not found to be statistically significant (*p* = 0.18). Central retinal thickness and macular volume did not show significant changes after VN treatment, with baseline values of 187 (±33) μm and 3.22 (±2.43) mm³, respectively, compared to 177 (±28) μm and 3.04 (±2.97) mm³ at the last follow-up (*p* = 0.34, *p* = 0.39). These results can be seen in Table 2 and Figure 1.

Retinal atrophy was defined as the development of new atrophy post gene therapy. All subjects developed some retinal atrophy at the retinotomy (injection) site. A total of 10 out of 12 eyes also developed some retinal atrophy away from the areas of injection. In eight eyes, the atrophic changes were subtle, asymmetrical and were located within the area of the bleb. In a young patient (P3) who had substantial and consistent improvements in FST, bilateral symmetrical retinal atrophic changes also developed outside the bleb area and in addition to atrophic changes within the bleb area (Figure 2). In the same patient, foveal retinal atrophy developed in the right eye (Figure 3)—without any loss in BCVA and without an obvious impact on the visual field at 12 months. In a 44-year-old patient (P6), retinal atrophy was observed as early as one week after the gene therapy. Atrophy originated at the retinotomy and extended inferiorly (Figure 4).

Three patients (P4, P5, P6) (six eyes) developed an elevation in IOP. Initially, all three patients were treated with acetazolamide 250 mg modified-release capsules twice daily along with IOP-lowering drops. In the case of P6, the IOP increase was temporary and effectively managed using the aforementioned therapy. For P4 and P5 (four eyes), surgical intervention for glaucoma was necessary, involving either goniotomy or tube surgery in order to control the elevated IOP. In P5, the elevated IOP was successfully controlled with goniotomy in both eyes, six months after the gene therapy, without the patient displaying any signs of glaucomatous changes. However, in the case of P4, even though tube surgery in both eyes normalized IOP levels, glaucomatous optic disc changes were observed. P1 (both eyes) developed mild intraocular inflammation one month following gene therapy with 1+ cells in the anterior chamber. This inflammatory response was effectively managed with an extended course of topical steroid treatment. Visually significant cataracts with corresponding vision worsening were observed in two patients (P4, P5) (four eyes) following the gene therapy. Cataract surgery was not performed during the study period. None of our patients developed subretinal deposits.

## 4. Discussion

In our study, we present the functional, morphological outcomes and adverse effects of VN treatment in 12 eyes from a cohort of six patients treated at the Oxford Eye Hospital. We confirmed the efficacy of treatment regarding overall FST improvements. These improvements were primarily observed in younger patients, whilst older patients with more advanced disease and fewer remaining viable retinal cells showed less improvement. We confirmed retinal atrophy as an important adverse effect, as described in other reports, and also described retinal atrophy affecting the fovea. 

Patients with *RPE65-*associated IRDs often present with severe vision impairment or blindness, nystagmus and night blindness from an early age [24]. Therefore, photoreceptor degeneration, amblyopia and nystagmus are considered limiting factors for BCVA improvements despite VN treatment [11]. Although a slight BCVA improvement was observed in the pivotal study [5] and meta-analysis [25], no statistically significant difference was observed after VN treatment in the previous studies, which is in concordance with our study. VN treatment improvements in *RPE65-*associated IRDs are age-dependent [26], which was confirmed by significant BCVA improvements in studies with pediatric patients, with the oldest patient treated at 16 years of age [8,11]. However, it is difficult to determine whether an improvement in visual acuity in pediatric patients is due to direct effects on cone function, a reduction in nystagmus or both. A study showed that VN treatment in biallelic *RPE65* IRDs markedly reduced the amplitude of infantile pendular nystagmus [8,11], which is known to be associated with improved BCVA [27]. Therefore, BCVA might be underestimated in these patients prior to gene therapy. In our study, there was a slight worsening in BCVA, but this change was not statistically significant. The slight deterioration in BCVA was most likely associated with cataract development in four eyes (P4, P5) and glaucomatous changes in two eyes (P5). 

FST is a fixation-independent light sensitivity test that allows for the evaluation of visual function in patients without stable fixation, very poor BCVA and a limited visual field [28]. Since it evaluates the lowest illumination perceived throughout the entire visual field when performed following dark adaptation, it is unaffected by the presence of nystagmus, but lacks spatial resolution [5]. Significant and rapid improvements in mean FST (white light) were observed within one month in the pivotal study [5] and real-world studies [21,29]. A review article highlighted sustained FST improvements in patients after VN treatment for up to 7.5 years [30]. In our cohort, the mean FST (white) also showed improvement in line with previous reports. The lack of statistical significance is likely due to the low number of participants and the baseline characteristics of our patient cohort. Indeed, significant improvements in FST (at least −16.5 dB improvement) were observed in all four eyes of the two younger patients (P1, P3). In contrast, FST improvements were only minimal in the eyes of patients aged 40 years or above (P2, P4, P5, P6). Limited FST improvements in older patients within our cohort may, in part, be due to cataract development in two patients and glaucomatous optic nerve changes in one patient. 

Regarding the morphological changes observed on OCT in our cohort, there was a slight decrease in macular volume and central retinal thickness following VN treatment. However, these changes did not reach statistical significance. A retrospective study of 27 eyes demonstrated a small, yet statistically significant, reduction in central retinal thickness following VN treatment [29]. In a case report, it was observed that after VN treatment, there was a swift improvement in foveal morphology, characterized by a distinct signal attributed to the external limiting membrane on OCT and photoreceptors, as visualized via adaptive optics retinal imaging [13]. Despite observing morphological improvement on multimodal imaging, the central retinal thickness did not change significantly following VN treatment [13]. In a retrospective review of six pediatric patients who underwent VN treatment, notable improvements in BCVA were observed at the 6-month visit. Although no significant changes were found in central retinal thickness at 6 months, there was a significant increase in the outer nuclear layer thickness [8]. This suggests that outer nuclear layer changes may serve as a better morphological biomarker for successful treatment outcomes than central retinal thickness. A study corroborated this finding by highlighting that assessing the entire retinal thickness is misleading, as it may be masked by the effects of retinal remodeling [31].

The development of retinal atrophy following VN treatment was initially reported by Gange et al. in 2022 [18]. It was defined as progressively enlarging retinal atrophy beyond the retinotomy site [18]. It was observed within and outside the area of the bleb in 55% of patients, and spared the fovea in all patients [18]. Despite the development of retinal atrophy, patients demonstrated consistent improvements in FST and GVF; however, 23.1% of patients developed paracentral visual field loss associated with the atrophy [18]. Subsequently, another retrospective study described retinal atrophic changes in all 13 eyes that underwent VN treatment [10]. Atrophy, which was preceded by AF changes, could be observed as early as two weeks after VN treatment. This study further validated that functional benefits (BCVA, FST, VF) were sustained despite the development of retinal atrophy [10]. Several theories have been proposed as to the mechanism of retinal atrophy development including: direct toxicity of the AAV2 vector associated with vector concentration and the CAG promoter; clinical and/or subclinical inflammation associated with gene therapy; or factors related to the technique of surgical delivery, including the size of the cannula, speed of injection rate, or possible creation of a pre-bleb [10,18]. In our cohort of patients, retinal atrophy developed in 10 out of 12 eyes treated with VN. Considering the high prevalence of retinal atrophy reported in recent studies, including our own cohort, it is surprising that no retinal atrophic changes were reported in the pivotal studies. It is not clear why the early clinical studies did not report the same adverse effect rates for retinal atrophy and IOP changes. This may reflect detection bias by the investigators or simply a chance result given the low number of patients treated in these trials. The pivotal phase 3 trial for VN included only 21 patients in the intervention group and 10 in the control group and was thus significantly underpowered to robustly report on the frequency of adverse effects [32]. Gange et al. reported that retinal atrophy developed in the perifoveal area, while the fovea itself remained unaffected [18]. We present a single case with retinal atrophy involving the fovea (P3, Figure 3). The foveal atrophy did not significantly affect the patient’s subjective and objective visual function (including BCVA). This may seem somewhat surprising, but can be explained by the pre-existing amblyopia, nystagmus and poor fixation due to poor cone-mediated vision from birth. 

Atrophy in this patient developed gradually and extensively outside the area of the subretinal bleb (Figure 2). This observation indicates that slower contributing factors, such as subclinical inflammation, may play a role in the development of retinal atrophy in this patient, who had large improvements in FST in both eyes (around −34 dB). In contrast, another young patient (P1) in our study demonstrated more moderate yet significant improvements in FST in both eyes, with gains of approximately −18 dB in both eyes. Notably, this patient did not exhibit any signs of retinal atrophy. These observations lend themselves to the speculation that there may be an optimal range of moderate FST improvements where the risk of significant retinal atrophy development is relatively low. On the other hand, the patient in Figure 4 (P6) developed a rapid onset of retinal atrophy, within the area of the subretinal bleb. Therefore, we postulate that early-onset retinal atrophy may be associated with the subretinal injection procedure itself, while late-onset retinal atrophy may be linked to subclinical inflammation.

In the pivotal study, it was observed that 20% of patients experienced a mild and transient increase in intraocular pressure (IOP), which was successfully treated with topical therapy [5]. This incidence rate of elevated IOP is similar to that reported in eyes following vitrectomy (24.2%) [33]. Therefore, it is likely that the IOP elevation observed in the pivotal study was associated with vitrectomy and subretinal injection. In contrast, our study identified two brothers (P4, P5) (four eyes) who developed sustained elevation in IOP, necessitating glaucoma surgery for effective IOP control. These patients had received topical and perioperative oral steroids as per the protocol. In addition, Kenalog was used to stain vitreous remnants during surgery, which adds to the steroid load of the eye by contrast to the pivotal studies, where no staining agent for vitreous remnants was used (personal communication with Dr. Russell). Previous studies have demonstrated that patients with steroid-induced glaucoma may experience persistently elevated IOP for several months, with approximately 26.5% of patients ultimately requiring surgical intervention [34]. Therefore, we postulate that the sustained IOP elevation observed in our two patients was a consequence of steroid-induced raised IOP presumed to be due to the Kenalog, which was successfully lowered by glaucoma surgery. 

The limitations of our study include its retrospective design, a relatively small sample size, and the absence of a control group.

It is of great interest to all stakeholders—patients, doctors, payers, industry and regulators—that the latest evidence from real-world studies is used to further refine patient selection for VN therapy. Currently, the criteria for offering treatment with VN to patients is based on a clinical diagnosis of an IRD with confirmed biallelic mutations in the gene *RPE65* and evidence of enough viable retinal cells. The latter is not well defined, leading to variable interpretations even among experts. As a result, there is a lack of consensus among ocular gene therapy centres regarding the most suitable candidates for treatment. This highlights the need to conduct further studies aimed at identifying clinical, functional or morphological biomarkers that can predict a favorable response to VN treatment. In conclusion, our findings highlight significant functional improvements observed following VN treatment in younger patients, while also noting the limited improvement observed in patients with more advanced disease and fewer remaining viable retinal cells. However, stabilization may still be a valuable quality-of-life improvement for patients. Retinal atrophy, which was observed in the majority of patients and was not associated with loss of visual function, may be more prevalent than previously reported. Further studies should focus on defining potential biomarkers such as age, baseline FST and outer nuclear layer thickness that would predict a good response to VN treatment, to aid in the selection of appropriate candidates for the therapy. More careful monitoring may be required in order to aggressively treat high IOP in patients who respond to steroids and reduce the risks of further complications. Therefore, our results showed that VN treatment remains an important and effective therapy for *RPE65*-associated IRDs. It notably improves visual function in patients, and thus, improves their quality of life. Although retinal atrophy is a significant adverse effect following VN treatment, it was not associated with a loss of visual function in our patients.

## Figures and Tables

**Figure 1 biomolecules-13-01484-f001:**
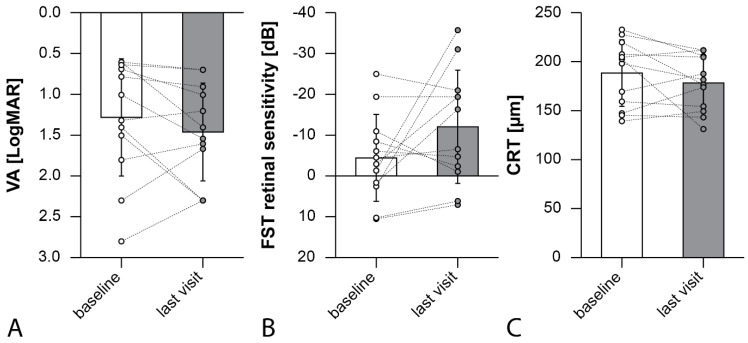
(**A**) Mean best-corrected visual acuity (BCVA) (*p* = 0.25), (**B**) mean Full-Field Stimulus Testing (FST, more negative values represent a higher threshold) (*p* = 0.18), (**C**) central retinal thickness (CRT) (*p* = 0.34) changes from baseline to the last visit. Each datapoint represents an eye.

**Figure 2 biomolecules-13-01484-f002:**
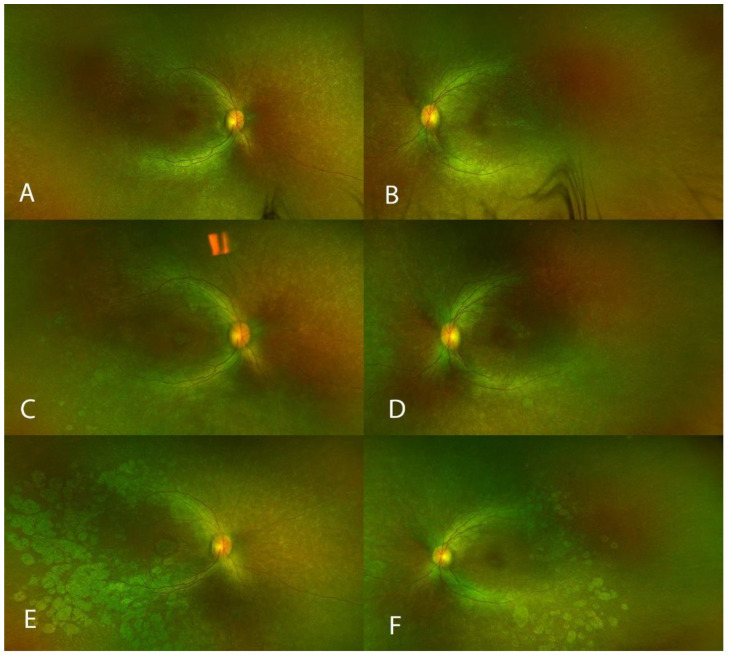
Development of retinal atrophy (Patient 3). Pseudo-color images showing the right eye (RE) (**A**) and the left eye (LE) (**B**) prior to gene therapy, at 6 months after the surgery ((RE) (**C**), (LE) (**D**)), and at 12 months after the surgery ((RE) (**E**), (LE) (**F**)).

**Figure 3 biomolecules-13-01484-f003:**
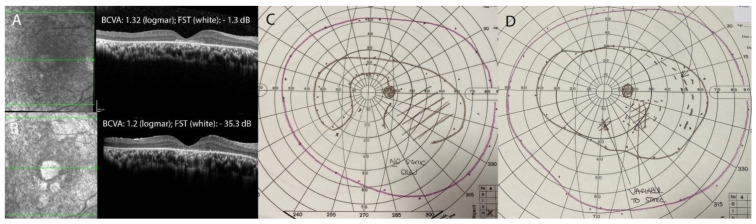
Worsening of fovea-involving retinal atrophy (Patient 3). Optical coherence tomography images showing the fovea prior to gene therapy (**A**) and at the 12-month follow-up (**B**). Goldmann visual field in the same patient prior to gene therapy (**C**) and at the 12 months follow-up (**D**); best-corrected visual acuity (BCVA); Full-Field Stimulus Testing (FST).

**Figure 4 biomolecules-13-01484-f004:**
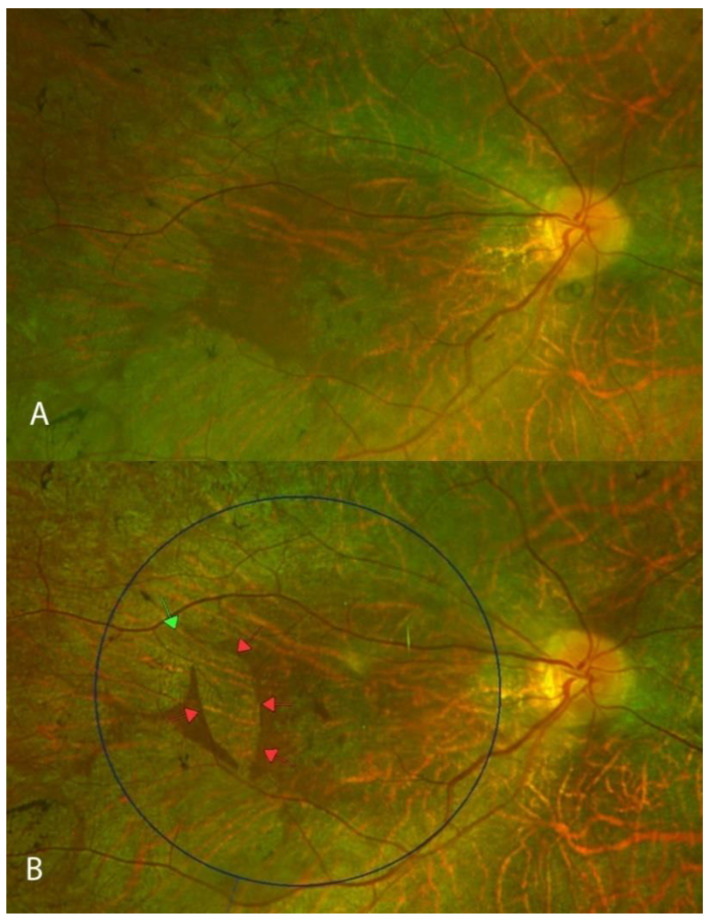
Development of retinal atrophy 1 week after gene therapy (Patient 6). Pseudo-color images prior to gene therapy (**A**) and one week after (**B**). The green arrow indicates the touch-down site of the subretinal cannula, the red arrows represent the area of new retinal atrophy, and the blue circle indicates the area of the raised bleb.

**Table 1 biomolecules-13-01484-t001:** Demographics and molecular genetic analysis in treated patients with biallelic *RPE65* mutations.

Age at Surgery, Sex, Ethnicity	Clinical Diagnosis	*RPE65* Mutation(s)	Gene Mutation Classification	Functional Changes
P1, 18 years, Male, Black British	LCA2	Compound Heterozygous *RPE65* C.271C > T; *RPE65* C.1102T > C	Missense; Missense	p.(Arg91Trp); p.(Tyr368His)
P2, 49 years, Male, White	RP20	Compound Heterozygous *RPE65* C.11 + 5G > A; *RPE65* C.1543C > T	Intronic; Missense	Disruption of normal splicing; p.(Arg515Trp)
P3, 19 years, female, Arabic	LCA2	Homozygous *RPE65* C.271C > T	Missense	p.(Arg91Trp)
P4, 40 years, Male, Pakistani	RP20	Homozygous *RPE65* C.179T > C	Missense	p.(Leu60Pro)
P5, 48 years, Male, Pakistani	RP20	Homozygous *RPE65* C.179T > C	Missense	p.(Leu60Pro)
P6, 44 years, Male, White	RP20	Homozygous *RPE65* C.560G > A	Missense	p.(Gly187Glu)

Retinitis pigmentosa (RP), Leber congenital amaurosis type 2 (LCA2).

**Table 2 biomolecules-13-01484-t002:** Functional and morphological outcomes in patients treated with voretigene neparvovec (VN).

	Prior to Gene Therapy			At the Last Follow-up			
	BCVA logMAR	FST (White) dB	CRT μm	BCVA logMAR	FST (White) dB	CRT μm	Follow-up Months
P1, 18 years, right eye	0.64	−2.9	227	0.69	−19.4	211	12
P1, 18 years, left eye *	0.78	1.9	219	0.9	−16.25	176	12
P2, 49 years, right eye *	1.8	−6	147	1.6	−4.7	174	12
P2, 49 years, left eye	2.8	−4.5	145	2.3	−6.5	143	12
P3, 19 years, right eye *	1.32	−1.3	139	1.2	−35.53	149	12
P3, 19 years, left eye	0.6	2.53	159	1.4	−30.95	154	12
P4, 40 years, right eye	0.6	−10.9	202	0.69	−1.2	131	6
P4, 40 years, left eye *	0.69	−8.4	207	1	−2.35	205	6
P5, 48 years, right eye *	1.5	10.35	198	2.3	6.15	181	6
P5, 48 years, left eye	1.4	10.5	169	2.3	7.06	187	6
P6, 44 years, right eye *	2.3	−19.3	232	1.66	−19.3	206	1
P6, 44 years, left eye	1	−24.9	204	1.54	−20.9	211	1

Best-corrected visual acuity (BCVA), Full-Field Stimulus Testing (FST), central retinal thickness (CRT), first eye treated (*).

## Data Availability

The data presented in this study are available on request from the corresponding author.

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
