# Peer review of "Outcomes and Adverse Effects of Voretigene Neparvovec Treatment for Biallelic RPE65-Mediated Inherited Retinal Dystrophies in a Cohort of Patients from a Single Center"

_biomolecules, 2023, doi:10.3390/biom13101484_

Round 1

Reviewer 1 Report

Kiraly et al. realized a very interesting article describing the “Outcomes and adverse effects of voretigene neparvovec treatment for biallelic RPE65-mediated inherited retinal dystrophies in a cohort of patients from a single centre”. This retrospective case series examines the functional and anatomical outcomes as well as adverse effects of voretigene neparvovec gene therapy in 6 patients (12 eyes) with biallelic RPE65-mediated inherited retinal dystrophy. The topic is highly relevant given the recent approval of voretigene as the first gene therapy for an inherited retinal disease. The study methods are appropriate, and the results contribute useful real-world data from a moderately sized cohort. However, the manuscript would benefit from some revisions to strengthen the presentation and interpretation of the findings.

Specific Comments:

  • The introduction provides good background on voretigene neparvovec and summarizes previous study findings well. Consider expanding on the rationale and importance of real-world evidence studies like this one.
  • In the methods, please provide more details on the inclusion/exclusion criteria, how patients were selected for treatment, and the procedure for obtaining informed consent.
  • The results are presented clearly but should highlight the key significant and non-significant findings. For example, annotate the p-values for the comparisons in visual acuity, FST, etc.
  • In the discussion, compare your efficacy results to previous studies in more depth. Your adverse effect rates for retinal atrophy and IOP changes seem higher than previously reported – expand on potential reasons.
  • Discuss limitations including the retrospective design, small sample size, and lack of control group.

·       Furthermore, I suggest adding data related to recent bulk transcriptomics studies which could represent a strong substrate to enforce the role of described molecular mechanisms, such as the recent PMID: 27737651, PMID: 26115622 and PMID: 32184807.

  • The conclusion could be strengthened by providing more specific recommendations or implications based on your findings.

Minor Comments:

  • Define abbreviations like BCVA, FST, etc. at first use.
  • Carefully proofread the manuscript to fix minor grammar, style issues.
  • Double check that all cited studies are included in the references list.

Overall, this is a well-conducted retrospective case series that provides useful real-world outcome data after voretigene treatment. Addressing the above comments would further improve the quality of the manuscript.

The English should benefit of an overall improvement.

Reviewer 2 Report

This  study evaluated morphological and functional outcomes, and side effects of voretigene neparvovec (VN) gene therapy for RPE65-mediated inherited retinal dystrophies (IRD) in 12  eyes (6 patients). Severe adverse effect was detected including retinal atrophy. Here are several comments:

1. All patients underwent molecular genetic analysis, which confirmed the presence of biallelic RPE65 mutations. Are those patients screened for other RP and LCA mutations? this screening seems important while most patient showing adverse effect  from treatment, the additional genetic screeening results should be provided to exclude any other unwanted disease mutations which could interfer with treatment. 

2.  ethnicity of those 6 patients should be cleary presented. if result is asscoiated with  ethnicity, it should be discussed.

3. it is said that all patients underwent a multidisciplinary team  review to confirm indication of treatment. Is there any standard  quantitative approach confirming that each patient has enough retinal cells to positively respond to treatment ?

Reviewer 3 Report

General comments

In this work the efficacy of voretigene neparvovec (VN; Luxturna), administered by subretinal injection, is assessed on 6 patients suffering from RPE65-associated IRD (namely RP or LCA). The study is well conducted and the techniques used to assess the functional and morphological outcomes (and adverse effects) of VN treatment are considered by this reviewer extensive and adequate. Also, despite the results are mostly negative (meaning not very promising or hopeful towards slowing RP/LCA progression), they are widely discussed and new research directions are proposed. Also unfortunately, the average BCVA remained roughly unchanged in all patients after VN surgery, for the reasons discussed by the authors. In addition this work points out several markers that could be predictive regarding the outcome of VN therapy and useful for the selection of candidates, such as ONL (better than whole-retina) thickness, age and (baseline) FST. Finally, although the number of patients investigated is reduced, results and conclusions regarding retinal atrophy development and adverse effects of VN administration are sound enough as to conclude that, in its current state-of-the-art, this treatment cannot be applied to the clinic. Having said all this, the following criticism is made by this referee in order to improve the quality and take-home messages of this paper.

Major points/concerns

Should the authors have used as a control the second, intact eye, to inject the empty AAV2 vector? Please state (briefly) why not done, and whether (instead) this is commonly done in other reports.

Usually one would expect that foveal atrophy is related to a loss of BCVA. If not so, please explain more clearly how/why this was not the case for patient P3 (Fig. 3, p. 7).

The Discussion section could, and should, be reduced in its length by ca. 25%.

Despite 10/12 (83.3%) of patients experiencing retinal atrophy, presumably as a consequence of the subretinal-injection treatment itself (or from subclinical inflammation), do the authors still believe this technique would be generally appliable to IRD (RP/LCA) patients in the near future? In addition to retinal atrophy, there are a good number of adverse effects that cannot be disregarded or easily circumvented, such as increased IOP, cataracts, intraocular inflammation, macular changes, etc. Could VN surgery still be useful in the clinic to retard RP/LCA progression? Could all current ‘troubleshooting’ be circumvented in a matter of years? What is the authors’ bet? What is its translational potential in the authors’ view? Please discuss briefly and draw one/two bottomline conclusions in this regard in the last paragraph of Discussion.

Minor, but not unimportant, points

Were subretinal deposits observed in any of the 6 patients who underwent VN gene therapy? Please state/discuss.

The Methods section is suggested to be divided into subsections and to provide a title to each of them.

Please correct all instances in which a number is directly followed by a unit without including a space in between. For example, ‘40mg/d’ (line 118), ‘20mg/d’ (lines 119-120) and ‘250mg’ (line 203).

Please write ‘second’ in full in line 120.

Replace ‘covid pandemic’ by ‘COVID-19 pandemic’ in line 124.

The word ‘and’ is missing between ‘surgery, and ‘post-surgery’ in line 125.

Authors are encouraged to designate patients as P1-P6 at all instances in which they cite them, and avoid other denominations. This applies, e.g., to ‘young patient’ (line 195; is this patient P1 or P3?) and to ’44-year-old-patent’ (line 199; who must be P6).

This study has been conducted on 6 patients (i.e., 12 eyes). Please correct the mistake in lines 300-301.

In Table 1, in the same fashion ‘LCA2’ is used for patients P1 and P3, ‘RP20’ must be used for the rest of patients (unless any of them exhibits choroidal involvement, in which case ‘RP87’ must be used instead).

Round 2

Reviewer 2 Report

recommend for publication.

minor improvement is needed.

Reviewer 3 Report

The article is deemed acceptable in present form.